# Chitosan Films Loaded with Alginate Nanoparticles for Gentamicin Release on Demand

**DOI:** 10.3390/polym17162261

**Published:** 2025-08-21

**Authors:** Cecilia Zorzi Bueno, Helton José Wiggers, Pascale Chevallier, Francesco Copes, Diego Mantovani

**Affiliations:** 1Laboratory for Biomaterials and Bioengineering (LBB-BPK), Associação de Ensino, Pesquisa e Extensão BI-OPARK, Max Planck Avenue, 3797, Building Charles Darwin, Toledo 85919-899, PR, Brazil; helton.wiggers@bpkedu.com.br; 2Laboratory for Biomaterials and Bioengineering (LBB-UL), Department of Mining, Metallurgical and Materials Engineering, CHU de Québec Research Center, Division of Regenerative Medicine, Laval University, Quebec City, QC G1V0A6, Canada; pascale.chevallier@crchudequebec.ulaval.ca (P.C.); francesco.copes.1@ulaval.ca (F.C.)

**Keywords:** chitosan, alginate, nanoparticles, antimicrobial, wound dressing

## Abstract

If untreated, skin wounds can lead to severe complications. Depending on the type of injury, long-term antibiotic administration is often required, and this decreases patient compliance. This limitation could be addressed by applying dressings capable of preventing infections by controlling drug release to the wound site. In this research, biodegradable wound dressings were investigated, based on natural polymers chitosan and alginate and incorporating the broad-spectrum gentamicin as antibiotic. Specifically, gentamicin was loaded into alginate nanoparticles, which were then loaded into chitosan-based films. This approach aimed at obtaining a system capable of modulating antibiotic release. The obtained nanoparticles had an average diameter of 86 nm and polydispersity index of 0.15. Antibiotic loading was around 600 µg/mg, with loading efficiency close to 100%. Films incorporating nanoparticles were compared to control films, which contained only gentamicin. Results showed that nanoparticles incorporation decreased film’s swelling in phosphate buffer saline, thus leading to a decrease in burst release while cytocompatibility for human dermal fibroblasts was maintained. Antibacterial activity was confirmed against both gram-positive and gram-negative bacteria. Moreover, the antibiotic was released as a function of pH, with distinct behavior at pHs ranging from 7.4 to 5.5. This indicates that alginate nanoparticles dispersed in chitosan films effectively release gentamicin on demand.

## 1. Introduction

Skin is the largest organ and the first barrier of the human body against the external environment. Due to its constant exposure, the skin is continuously subjected to wounding, which in turn is quite prone to inflammation or infection that slows down the healing process and poses a substantial risk of inflicting severe damage to tissues and cells [1,2]. The topical use of antibiotics remains the most common method for clinical treatment of wound infections. However, the misuse of antibiotics can contribute to drug resistance, complicating and delaying wound healing [3].

In this sense, smart wound dressings for infected wound treatment are being investigated worldwide. Ideally, these dressings should be biocompatible, biodegradable, capable of promoting tissue regeneration and preventing or inhibiting bacterial infection through controlled drug release [3,4]. Stimuli-responsive dressings, such as the ones capable of releasing drugs by pH and temperature changes, are some examples of these smart dressings [2,5].

Natural polymers represent one of the best options to produce wound dressings. Among them, chitosan and alginate are particularly interesting due to their rather recognized properties. Chitosan is obtained by deacetylation of chitin, which is the second-most abundant polysaccharide in the world. Due to their good biocompatibility, low cytotoxicity, controllable biodegradability, stimuli-responsive properties and reasonable cost, chitosan-based materials have been widely used in drug delivery [3,6]. Nevertheless, the mechanical properties of chitosan as well as its solubility in water represents a challenge for several applications [6]. Alginate is also a biocompatible polysaccharide, and it is mainly obtained from brown algae. Alginate finds many applications in food and drug delivery, as it forms smart hydrogel materials capable of responding to different stimuli [7].

Recent advances in polymeric wound dressings have focused on combining natural polymers with active agents to promote bioactivity through drug release and with crosslinkers, plasticizers or other compounds to control physicochemical properties and drug-release profiles [8,9]. Active agents such as essential oils are widely studied [10], yet issues of volatility, instability and hydrophobicity remain a challenge [11]. The combination with nanostructured systems is also studied as a strategy to enhance the dressing’s efficacy, control drug release and preserve biocompatibility. Several studies have incorporated antimicrobial metallic nanoparticles (e.g., Ag, ZnO, TiO_2_) into polymeric hydrogels; however, there is the risk of deposition and toxic effects in organs such as the skin, liver and kidneys with prolonged usage [12].

An emerging and high-potential approach to control drug release is to combine drug-loaded nanoparticles and polymeric matrices, which was reported to prevent initial burst, improve the stability of drugs and reduce toxic side effects [1,13]. This study is based on this approach, and it was implemented in two steps. First, gentamicin-loaded alginate nanoparticles were developed using a simple and reproducible method. Second, a double encapsulation system was designed by embedding the nanoparticles into a chitosan-based matrix crosslinked with tannic acid–iron (Fe^2+^) complexes [14]. To our knowledge, this is one of the first studies reporting the development and the characterization of a pH-sensitive composite system using alginate–chitosan combinations for the delivery of gentamicin on demand. The physicochemical and biological properties of the newly developed nanoparticles and those of the resulting films were characterized, and their pH responsiveness was evaluated.

## 2. Materials and Methods

### 2.1. Materials

Low-viscosity sodium alginate with a molecular weight estimated around 12–40 kg/mol (W201502, Sigma-Aldrich, Wuxi, China) [15], chitosan with an average molecular weight of 217 kg/mol, polydispersity index of 1.3 and deacetylation degree of 82.2% [16] (448877, Sigma-Aldrich, Shanghai, China), tannic acid ACS (Sigma-Aldrich, Wuxi, China), iron sulphate heptahydrate (Êxodo Científica, Sumaré, Brazil), gentamicin sulfate (Sigma-Aldrich, St. Louis, MO, USA), acetic acid 99.7% (Synth, Diadema, Brazil), phosphate-buffered saline (PBS) (Sigma, Gillingham, UK), Mueller–Hinton agar (Kasvi, Madrid, Spain), acetonitrile > 99.9% (Merck, Darmstadt, Germany), formic acid ≥ 95% (Synth, Diadema, Brazil), trifluoroacetic acid ≥ 99.5% (Sharlau, Barcelona, Spain), *Escherichia coli* ATCC 8739 (Lab-Elite™, St. Cloud, MN, USA), *Staphylococcus aureus* ATCC 6538 (Lab-Elite™, Cloud, MN, USA), glycerol ≥ 99.8% (Neon, Suzano, Brazil), human dermal fibroblasts C0045C (Gibco, Invitrogen, Burlington, ON, Canada), Dulbecco’s modified Eagle’s medium (DMEM) (Gibco, Invitrogen, Burlington, ON, Canada), fetal bovine serum (FBS) (Gibco, Invitrogen, Burlington, ON, Canada), penicillin (Gibco, Invitrogen, Burlington, ON, Canada), streptomycin (Gibco, Invitrogen, Burlington, ON, Canada), trypsin (Gibco, Invitrogen, Burlington, ON, Canada) and resazurin sodium salt (Sigma-Aldrich, Burlington, ON, Canada) reagents were used as received, without further purification.

### 2.2. Nanoparticles Preparation

Alginate–gentamicin nanoparticles (NPs) were prepared by the dropwise addition of a gentamicin sulphate (GS) solution at 5 mg/mL to 20 mL of an alginate (A) solution at 0.2% *w*/*v*. The solutions were mixed under magnetic stirring at 1500 rpm for 5 min, followed by 20 min centrifugation at 3500 rpm to remove aggregates. The GS/A ratio varied from 30 to 60% *w*/*w*. This range was chosen since ratios below 30% did not enable nanoparticles formation and ratios above 60% led to precipitation of large particles.

### 2.3. Films Preparation

Chitosan-based films loaded with NPs were prepared based on a previous publication [14]. Chitosan (CS) was dissolved in acetic acid 1% *v*/*v* at 1.5% *w*/*v*, and tannic acid (TA) and iron sulfate (Fe) were dissolved in ultrapure water at 50 mg/mL and 3 mg/mL, respectively. Firstly, 6.67 g of CS solution were weighed, followed by the addition of 1.67 mL of Fe and 0.4 mL of TA, under magnetic stirring. NPs were added at a final antibiotic percentage of 9% (*w*/*w*) in relation to CS. The mixture was then placed in a 90 mm Petri dish, and films were obtained by casting at 37 °C. Films containing only the antibiotic at 9% (*w*/*w*) were also prepared and used as controls.

### 2.4. Gentamicin Quantification

Gentamicin quantification in NPs and films was performed using high-performance liquid chromatography coupled to mass spectrometry (HPLC–MS, Waters, Milford, CT, USA). Chromatographic separation was achieved with a Triart C18 column 250 mm × 4.6 mm, 3 μm (YMC, Kyoto, Japan), at 30 °C. The mobile phase consisted of water/acetonitrile (20:80 *v*/*v*), 0.5% (*v*/*v*) trifluoroacetic acid and 0.5% (*v*/*v*) formic acid with a constant flow rate of 0.5 mL/min. The HPLC Waters 2696 system was connected to a Micromass Quattro Micro API with a multimode source. Electrospray ionization (ESI) in positive mode was used with a gas temperature of 350 °C and a vaporizer temperature of 150 °C. Nitrogen was the drying gas at a flow rate of 400 L/h. The capillary voltage was 3000 V, with a multiple reaction monitoring dwell time of 600 ms, fragmentation voltage of 50 V and collision energy of 10 V. Volumes of 20 μL of the thawed solutions were injected and compared to a standard gentamicin curve. The limits of quantification and detection used for this method were 0.105 and 0.031 μg/mL, respectively [14].

### 2.5. Nanoparticles Characterization

#### 2.5.1. Dynamic Light Scattering

NPs size, polydispersity index and zeta potential were assessed using dynamic light scattering (DLS) in a Zetasizer Nano ZS90 (Malvern^®^_,_ Worcestershire, UK) with a 4 mW 632.8 nm laser and 90° angle. The equipment was set with the following parameters: 3 runs with automatic number of repetitions and temperature of 25 °C. The results are presented as mean and standard deviation.

#### 2.5.2. Transmission Electron Microscopy

The morphology of the NPs was analyzed using transmission electron microscopy (TEM) in a JEOL JEM 1400 (Tokyo, Japan) with an acceleration voltage of 40 kV–120 kV and 2 nm end-to-end resolution. Samples were deposited onto 200 mesh formvar/carbon-supported copper grids previously glow discharged with air. The sizes of the NPs were measured using ImageJ software version 1.54f.

#### 2.5.3. Gentamicin Incorporation

Gentamicin incorporation into the NPs was evaluated by centrifuging the samples in Amicon filters (100 kDa) at 3500 rpm for 20 min. The collected filtrate was analyzed using HPLC–MS. The loading efficiency (*LE*) and loading capacity (*LC*) were calculated according to Equations (1) and (2), respectively:(1)$$LE\;=\;\frac{W_{T}-W_{R}}{W_{T}}\;\times \;100$$(2)$$LC=\frac{W_{T}-W_{R}}{W_{P}}$$
where *W_T_* is the total amount of antibiotic used for nanoparticles synthesis, *W_R_* is the amount of antibiotic in the filtrate and *W_P_* is the amount of polymer used for NPs synthesis. The experiments were performed in triplicate and the results are presented as mean and standard deviation.

#### 2.5.4. Gentamicin Release from Nanoparticles over Time

The samples were mixed with 1 × PBS (2 mg of polymer in 2 mL of PBS) and transferred to 100 kDa Amicon tubes. The mixtures were kept at 37 °C and 150 rpm. Time points were collected every 30 min up to 4 h. Before each time point collection, the tubes were centrifuged at 3500 rpm for 15 min, and the filtrate was separated for later quantification using HPLC–MS. After each time point collection, 2 mL of fresh PBS were added to the samples. The experiments were performed in triplicate and the results are presented as mean and standard deviation.

### 2.6. Films Characterization

#### 2.6.1. Morphology

The morphology of the films was assessed by visual inspection and by scanning electron microscopy (SEM). First, the samples were deposited on a tape and metalized with gold for 3 min. The analysis was carried out using a TESCAN VEGA3 microscope (TESCAN, Brno, Czech Republic), operating between 10 and 15 kV at a working distance of 7–8 mm. The images were taken at 500× and 7000× magnification.

#### 2.6.2. Thickness

Thickness was measured using a digital micrometer (Mitutoyo, Kawasaki, Japan) at five different positions of the films. The results are presented as mean and standard deviation.

#### 2.6.3. Swelling and Mass Loss

The films were cut into small squares of approximately 1 mg, which were immersed in 2 mL of PBS and incubated for 24 h at 37 °C and 150 rpm.

For swelling measurements, the samples were washed twice with distilled water and blotted with filter paper to remove excess water. The swollen samples were then weighed, and their swelling ratio (*SR*) was calculated according to Equation (3):(3)$$SR\;=\;\;\frac{W_{S}-W_{0}}{W_{0}}\times 100\%$$
where *W*_0_ is the initial weight and *W_S_* is the swollen weight.

For mass loss measurements, the samples were washed twice with distilled water and dried at 37 °C until constant mass was reached. The mass loss (*ML*) was determined according to Equation (4):(4)$$ML\;=\;\;\frac{W_{0}-W_{D}}{W_{0}}\times 100\%$$
where *W*_0_ is the initial weight and *W_D_* is the final dried weight.

For both swelling and mass loss, the experiments were carried out in quintuplicates, and the results are represented as the mean and standard deviation.

#### 2.6.4. Mechanical Properties

The mechanical properties of the Film Control and Film NPs were evaluated through tensile testing using a universal testing machine (HSensor, Maringá, Brazil), following a modified version of the ASTM D882 standard [17]. Films were cut into 7 cm × 1 cm samples and conditioned in a desiccator to eliminate moisture variability before running the tests. Afterwards, the thickness of the samples was measured from five points (as described in Section 2.6.2), and the average values were recorded. The analysis was performed with 10 replicates for each film, using a 5 N load cell at 5 mm/min, with an initial grip separation of 50 mm. Tensile strength (TS), elongation-at-break (EB) and Young’s modulus (YM) were obtained from the stress–strain graphic, where TS and EB are the maximum stress and elongation tolerated by the sample, respectively, and YM is the slope of the elastic region of the stress–strain graphic.

#### 2.6.5. Fourier Transformed Infrared Spectroscopy-Attenuated Total Reflectance (FTIR-ATR)

FTIR-ATR analysis of chitosan, alginate, Film Control and Film NPs was conducted in a PerkinElmer Frontier spectrometer (PerkinElmer, Shelton, CT, USA) in transmittance mode from 400 to 4000 cm^−1^, with 45 scans and a resolution of 4 cm^−1^. Each sample was measured in triplicate.

Chitosan and alginate samples were prepared as follows: 10 mL of a chitosan solution at 1.5% (*w*/*v*) in acetic acid at 1% (*v*/*v*) and 10 mL of an alginate solution at 2% (*w*/*v*) in water were casted separately in two 90 mm Petri dishes, and the obtained films were used for analysis.

#### 2.6.6. Differential Scanning Calorimetry (DSC)

DSC analysis of chitosan, alginate, Film Control and Film NPs was carried out using a differential scanning calorimeter (DSC; Shimadzu-DSC-60, Kyoto, Japan). Samples were precisely weighed (4 ± 0.25 mg), placed in aluminum pans and heated from 30 to 500 °C at 20 °C/min under constant nitrogen flow of 50 mL/min. The analysis was performed in triplicate. Chitosan and alginate samples were prepared as described in Section 2.6.5.

#### 2.6.7. Gentamicin Release from Films over Time

Film disks (15 mm diam.) were placed in polypropylene tubes. Then, 2 mL of PBS (pH 7.4) or 2-(N-morpholino) ethanesulfonic acid (MES) buffer (pH 5.5) was added into each tube, which were kept at 37 °C under agitation at 150 rpm. The release media were collected over time at 0 h, 1 h, 4 h, 24 h, 3 days, 7 days and every 7 days until the antibiotics were no longer quantifiable. After each collection, fresh buffer solution (PBS or MES) was added to the tubes. Antibiotic quantification was carried out using HPLC–MS (described in Section 2.4). Experiments were carried out in triplicate, and the results are presented as mean and standard deviation.

### 2.7. Antibacterial Assays

#### 2.7.1. Bacteria Stock Preparation

The experiments were performed for *Escherichia coli* (*E. coli*) and *Staphylococcus aureus* (*S. aureus*). Each bacterium was seeded on sterile Mueller–Hinton agar on Petri dishes and incubated in an inverted position overnight at 37 °C. A single colony was picked and placed in 20 mL of sterile Mueller–Hinton broth overnight at 37 °C under agitation at 150 rpm. Then, sterile glycerol at 20% was added, and bacteria aliquots were frozen at −20 °C.

#### 2.7.2. Antibacterial Activity

The Kirby–Bauer susceptibility test was used in this study [18]. Briefly, the bacteria to be tested were taken from frozen stocks, thawed to room temperature and quantified by a log dilution enumeration method [19]. Then, 300 μL containing approximately 1 × 10^8^ CFU/mL, was spread with a Drigalski spatula onto 15 cm Petri dishes coated with fresh sterile Mueller–Hinton agar.

The test was performed for both NP suspensions and for chitosan films. For the analysis of NPs, small holes were made in the agar with the aid with a 5 mL plastic syringe. The samples and antibiotic controls were then pipetted into the holes, resulting in a total amount of 10 μg of antibiotic per hole. For the analysis of the films, samples of 6 mm diameter [20] were cut and sterilized under UV light for 15 min on both sides. Afterwards, the disks were placed on the Petri dishes containing the bacteria. Paper disks impregnated with 10 μg of the antibiotic were used as positive control, and paper disks without antibiotic were used as negative control.

The Petri dishes containing the samples and the bacteria were incubated overnight at 37 °C. Afterwards, the inhibition zones were measured using a digital pachymeter in three different positions. Experiments were carried out at least in duplicate for each bacterium, and the results are presented as mean and standard deviation.

### 2.8. Biocompatibility

#### 2.8.1. Cell Culture

The effects of the films on cell viability were analyzed using human dermal fibroblasts (HDFs). The cells were cultured in Dulbecco’s modified Eagle’s medium (D-MEM) with 10% fetal bovine serum (FBS), penicillin (100 U/mL) and streptomycin (100 U/mL), at 37 °C, in a saturated atmosphere at 5% CO_2_. The culture medium was changed every 48 h until 85–90% of confluence was reached. Then, cells were enzymatically detached from the culture plates with 0.05% trypsin and reseeded at a ratio of 1:3 or used for experiments. Cells at passage 7 were used for the experiments.

#### 2.8.2. Indirect Cytotoxicity Test

An indirect cytotoxicity assay of the films was performed based on the procedure ISO 10993-5 [21]. Films samples were cut into 1 cm^2^ and sterilized with UV irradiation, undergoing 2 cycles of 15 min on each side. Afterwards, samples were immersed in 660 µL of D-MEM supplemented with 1% penicillin and 1% streptomycin and incubated at 37 °C in a saturated atmosphere at 5% CO_2_ for 24 h. After incubation, the media, referred to as extracts, were collected from the samples and supplemented with 10% FBS.

HDFs were seeded in the wells of 96-well microplates at a density of 20,000 cells/cm^2^ and incubated at 37 °C, 5% CO_2_ for 24 h in 100 µL/well of complete medium. The day after, the media were removed and 100 µL of the extracts were added to the wells containing cells. Normal HDFs complete medium was used as a control. After 24 h of incubation, the extracts were removed and 100 μL of 1 × resazurin sodium salt solution in complete medium was added to the cells, which were incubated for 4 h at 37 °C and 5% CO_2_. Then, the solutions containing the now-reduced resorufin product were collected, and fluorescence intensity at 545 nm_ex_/590 nm_em_ was measured with a SpectraMax i3x Multi-Mode Plate Reader (Molecular Devices, San Jose, CA, USA). Fluorescence intensity is proportional to cell viability. Experiments were performed in triplicate, and the results are presented as mean and standard deviation.

### 2.9. Statistical Analysis

All the obtained results are expressed as the average values of the experimental measurements and standard deviation (±SD). Data averages were analyzed with Statistica^®^ software version 8, by applying the Tukey test with a 5% (*p* < 0.05) significance level.

## 3. Results and Discussion

In this study, a straightforward method to produce antimicrobial nanoparticles of alginate loaded with gentamicin (NPs) was developed. The obtained NPs, prepared with different gentamicin proportions, were characterized by DLS, and the size, the polydispersity index (PDI) and the zeta potential were measured. Additionally, by HPLC–MS, the antibiotic loading capacity (LC) and loading efficiency (LE) were also assessed. Results are shown in Table 1.

As can be seen from Table 1, the variation in gentamicin proportion caused significant changes in the NPs. While size, PDI and NPs’ negative charge decreased with increasing gentamicin content, loading capacity increased significantly and loading efficiency increased slightly. Moreover, the increase in gentamicin content improved the reproducibility in the synthesis of the NPs, as can be noticed by the decrease in the standard deviations shown in Table 1.

The decrease in the NPs’ size with the increase in gentamicin content could be explained by the electrostatic interactions between the guluronate moieties of alginate and amino groups of gentamicin [22,23]. These results align with the research of Heriot et al., in which the authors noticed a significant compaction and shrinkage of alginate gels upon addition of gentamicin. The densification of the gels followed a gentamicin concentration-dependence pattern, particularly for alginates rich in guluronate groups [22]. Therefore, higher amounts of gentamicin could cause the compaction of the NPs’ structure and enhance nucleation during NP formation, potentially leading to higher antibiotic loading, reaching efficiencies close to 100%. The decrease in the negative charge with increasing gentamicin content is probably due to the antibiotic amino groups, which are positively charged.

Among these three NPs, the most promising is the one prepared with 60% (*w*/*w*) of gentamicin, given its uniform size distribution, better reproducibility and higher gentamicin loading. This formulation was selected for its detailed characterization. Figure 1 shows the results for morphology analysis by TEM at different magnifications of 8000 and 50,000×. It is possible to visualize the amorphous shape of the NPs, which seems to be formed by the agglomeration of smaller structures. The sizes of the NPs measured using ImageJ software (v. 1.54f) were in between 90 and 150 nm, corroborating the sizes measured by DLS.

Figure 2A depicts gentamicin release over time from the NPs up to 4 h, with a burst in the first 100 min, in which 65% of the loaded antibiotic was released. After 100 min, gentamicin was slowly released, reaching a maximum of 74% of the loaded antibiotic. Furthermore, according to Figure 2B, the NPs displayed antimicrobial activity against bacteria commonly found in infected wounds, *E. coli* and *S. aureus*, with inhibition halo diameters comparable to the positive controls. Since in both the controls and NPs the gentamicin amount was 10 µg, it appears that the gentamicin loaded into the NPs kept its full efficiency.

The developed nanoparticles were then loaded into chitosan-based films (Film NPs), which were prepared according to a previous study [14]. The obtained films were characterized regarding their morphology, thickness, swelling, mass loss in PBS, mechanical and thermal properties, and by FTIR-ATR. Control films loaded with gentamicin (Film Control) were also prepared and characterized for comparison.

The morphology of the films was assessed by visual inspection and by SEM, and the images are shown in Figure 3. While both films are dark brown, the presence of NPs resulted in darker and more opaque films than the control films (see Figure 3A). The micrographs show that the Film Control has a smooth and compact surface without pores or cracks (Figure 3B), which indicates good structural integrity due to the semi-crystalline nature of chitosan film obtained by dissolving the polymer in acetic acid [24,25,26]. Film NPs exhibit a rougher surface, attributed to the presence of nanoparticles (Figure 3C). The transversal section of the Film NPs clearly evidenced the presence of small white dots inside the film, attributed to the nanoparticles (Figure 3C).

The films’ thickness, swelling and mass loss in PBS and mechanical properties are shown in Table 2. Although thicknesses and mass loss values were similar, varying between 15 and 18 µm and between 22 and 25%, respectively, swelling percentage decreased significantly from 167 to 103% after nanoparticles incorporation. These results might be explained by the electrostatic interaction between alginate carboxyl groups and chitosan amino groups, increasing the film’s crosslinking degree. The observed variation in swelling behavior may have a direct impact on the applicability of the films and on the gentamicin release kinetics. According to Feng and Wang, hydrogels with swelling degrees higher than 150% have high wound exudate absorption capacity but poor stability and weakened bio-adhesion. On the other hand, hydrogels with swelling degrees lower than 150% have excellent dimensional stability, long-term wet-adhesion performance and persistent mechanical strength; however, they are not suitable for highly exuding wounds [27]. Mass loss values were consistent with a previous study, in which chitosan films crosslinked with tannic acid and iron sulphate had mass loss in PBS of approximately 23%, attributed to the release of non-trapped antibiotic and partial chitosan solubilization [14]. This partial degradation profile is advantageous for temporary wound coverage, where gradual material breakdown may reduce the need for dressing changes and support tissue remodeling.

Tensile strength values were of around 33 MPa for Film Control and 25 MPa for Film NPs, indicating that the incorporation of NPs into the films made them more brittle. On the other hand, elongation at break was similar for both samples, with values between 1.5% and 1.9%. Consequently, the Young’s modulus, which is calculated based on the slope of the stress-strain curve, was higher for the Film Control (38 MPa) compared to Film NPs (28 MPa). The decrease in mechanical resistance after NPs incorporation indicates a higher crosslinking degree due to the electrostatic interactions between chitosan and alginate, corroborating the results of swelling degree analysis. Commercial wound dressings usually present much lower tensile strength, from 0.04 to 4.6 MPa, and much higher elongation at break, from 27 to 814% [28]. Therefore, the developed films are more robust and can be considered suitable for use in low mobility wound areas where high flexibility is not required, or as protective outer layer in multilayered systems.

To further elucidate the structural and thermal characteristics of the films, FTIR-ATR and DSC analyses were carried out. The following results highlight the chemical interactions and thermal behavior that govern the stability and performance of the developed systems. The results are shown in Figure 4.

According to FTIR-ATR spectra of the films, the characteristic bands of chitosan, such as COC stretching vibrations at 1069 cm^−1^, amide II (NH(CO)) bond at 1545 cm^−1^ and amide I (C=O) bond at 1640 cm^−1^ [14], are predominant over those of the other components. Therefore, no significant difference was observed between Film Control and Film NPs. Nevertheless, a very subtle band at 831 cm^−1^, which is present only in Film NPs spectrum, could be attributed to a shift in the mannuronic acid residue of alginate, located at 815 cm^−1^ [29,30], indicating a possible electrostatic interaction between alginate and amino moieties of both chitosan and gentamicin. The spectra of both films differ from that of pure chitosan in some aspects, which could be due to the presence of the crosslinkers (tannic acid and iron) and the antibiotic (gentamicin). For example, the presence of tannic acid can be confirmed by the band at 1199 cm^−1^, attributed to C-O stretching, and by the band at 1707 cm^−1^, attributed to a shift of carbonyl groups stretching (1716 cm^−1^) after chelation with iron ions [14]. Moreover, the enlarged band at 1328 cm^−1^ is attributed to S=O stretching from sulfate moieties found both in Fe_2_SO_4_ and gentamicin [14].

According to DSC analysis, all samples had broad endothermic peaks around 105–110 °C, which can be attributed to water evaporation [31]. The pure polymers showed exothermic peaks indicative of thermal degradation at 246 °C for alginate and 310 °C for chitosan, which corroborates literature data [31,32]. The thermogram of the Film Control did not show the chitosan exothermic peak, which could possibly be attributed to interactions with other film components. Nevertheless, a small exothermic peak can be noticed at 265 °C, and it can be correlated to tannic acid thermal degradation [33]. Film NPs, on the other hand, showed an exothermic peak at 294 °C, which is between the characteristic peaks of chitosan and alginate, indicating the formation of a polyelectrolyte complex between these two polymers [31]. Therefore, DSC analysis was more elucidative than FTIR-ATR to identify possible interactions among the components of the films.

The results of films characterization by the Kirby–Bauer susceptibility test are shown in Figure 5. All samples presented antibacterial activity against *S. aureus* and *E. coli*, with halo diameters ranging from 21 to 25 mm. The values obtained for the paper disc impregnated with 10 µg of antibiotic, used as a control in the experiment, are consistent with those reported in the EUCAST Database. For *S. aureus*, Film NPs and Film Control showed statistically similar antibacterial activity. On the other hand, for *E. coli*, Film Control displayed a slightly higher antimicrobial activity. Although no clear trend was observed in the variation of inhibition halos between the tested samples, the preserved bactericidal efficacy of Film NPs indicates promising applicability of the films as antibacterial wound dressings.

The samples were also characterized in terms of their long-term release in two different pHs (5.5 and 7.4), according to the methodology described in 2.6.7. The released gentamicin percentages and concentrations against time (up to 14 days) are shown in Figure 6.

Film NPs and Film Control behaved differently according to pH. Film Control showed a slight increase of 8.5% in the release percentage at pH 5.5. On the other hand, Film NPs displayed a 50% increase in release percentage at pH 5.5 (Figure 6A). This interesting result means that Film NPs are pH responsive and release the antibiotic under acidic conditions.

The concentrations of gentamicin in the collected release media are also depicted in Figure 6B, where the dotted line represents gentamicin minimum inhibitory concentration (MIC) against *S. aureus* and *E. coli*, around 0.5 µg/mL. At pH 7.4, the gentamicin concentration falls below the MIC value after 7 days, indicating loss of antimicrobial activity. On the other hand, at pH 5.5, the antibiotic concentration is kept above the MIC value throughout the experiment for up to 14 days. Therefore, antimicrobial activity can be extended at pH 5.5. Given that most commercial dressings available on the market are recommended to be changed at a maximum of every 7 days [34], the developed films can be considered promising as wound dressings in the tested pH range.

The higher released amount of gentamicin at lower pH can be correlated to the protonation of chitosan (pKa ~ 6.5), which tends to dissolve in acidic conditions [6,35], inducing a decline in its interaction with other positively charged molecules, such as gentamicin (pKa 7.15–9.2) [36]. The higher pH sensitivity of Film NPs compared to Film Control is probably due to the presence of alginate, which is also pH sensitive (pKa 3.4–3.6) [7]. At pH 5.5, the alginate carboxyl groups become more protonated, decreasing their interactions with positively charged amino groups of gentamicin (pKa 7.15–9.2) [36] and chitosan (pKa~6.5). Moreover, the interactions between gentamicin and chitosan also decrease at pH 5.5. This combination of factors may facilitate antibiotic diffusion and release from the films at lower pHs.

pH responsiveness is of interest for potential wound dressing applications, which can adapt their functions to dynamic wound conditions [5]. According to Schneider et al. (2007), the pH value in wounds undergoes continuous alterations according to the healing phase, and it is also subjected to many different endogenous and exogenous factors [37]. Therefore, acute and chronic wounds present very different pH behaviors. In the initial healing phase, both acute and chronic wounds undergo an acidosis process, due to increased lactic acid and oxygen in the wound. However, while acute wounds undergo a normal healing process, restoring rapidly to the natural pH of the skin, chronic wounds tend to stay at an alkaline pH for long periods, which impairs healing and promotes bacterial growth [5,38]. In this sense, wound dressings that release an incorporated drug in response to acidic pHs are more adequate for wounds during the initial healing phase, what may prevent bacterial colonization and biofilm growth. Also, chronic wounds that are progressing in their healing process typically exhibit an acidic pH and can also benefit from acidic pH-responsive wound dressings [5].

Examples of chitosan-based dressings with response to acidic pH can be found in the literature. For instance, chitosan–tannic acid films loaded with neomycin released higher amounts of the drug at mildly acidic pH 5.5 compared to higher pHs. The authors attributed this result to chitosan dissolution in acidic media, allowing water to penetrate and solubilize the drug, which was released by diffusion. This behavior was considered favorable for the normal skin environment and the sustained release at pH 7.5 indicated potential for chronic wound infection [39]. In another example, chitosan and PNIPAm hydrogel films loaded with gentamicin sulphate were shown to be pH and temperature responsive, releasing more antibiotic at lower pH and higher temperatures. The authors observed that, in acidic solution, the amino groups of chitosan become protonated, causing repulsion between chitosan chains and releasing the antibiotic [40]. Other chitosan-based pH-responsive wound dressings were developed by blending chitosan, guar gum and polyvinyl alcohol for sustained paracetamol release. The hydrogels swelled the most in acidic pH, showing to be appropriate for controlled drug release [41]. In another example, regarding coatings for biomedical devices, chitosan–quercetin films loaded with trimethoprim released about 20% more antibiotic in pH 5.5 compared with pH 7.4, probably due to chitosan protonation and disruption of the intermolecular interactions with the antibiotic [42].

Other pH responsive wound dressings can be found in the literature, in which the drug release is based on other types of mechanisms, for example, a composite hydrogel consisting of chitosan, phenolic compound 3,4-dihydrocinnamic acid and curcumin-Fe^3+^ nanoparticles. Curcumin was released at low pH by the break of the coordination bond between curcumin and Fe^3+^ [43]. Moreover, a double network polyacrylamide catechol-chitosan hydrogel containing pH-responsive nanoparticles of acetalized cyclodextrin loaded with Resolvin E1, a pro-resolving lipid mediator, intelligently released Resolvin E1 at low pH. This hydrogel promoted the regulation of the differentiation of macrophages to the M2 phenotype [44].

It is also interesting to note that Film NPs have a lower burst release when compared to control films (Figure 6A). Burst reduction was more significative at pH 7.4, with a decrease of 57% of cumulative gentamicin release. One hypothesis for these results is that, at pH 7.4, alginate carboxylic groups are deprotonated and interact with positively charged amino groups from chitosan and gentamicin. Therefore, crosslinking degree increases and burst release is reduced. On the other hand, at pH 5.5, alginate can be protonated, facilitating gentamicin release, as stated before. Examples from the literature show that films or hydrogels incorporating drug-loaded nanoparticles have significant burst reduction, attributed to the limited drug diffusion caused by the nanoparticles [45,46,47]. For example, Buntum et al. (2021) observed, in poly(vinyl alcohol) hydrogels containing chitosan nanoparticles loaded with essential oils, a burst reduction of 35% for clove essential oil and 11% for turmeric essential oil [47].

The cytocompatibility of the produced films was also evaluated, and the results are shown in Figure 7. Both Film Control and Film NPs proved to be non-toxic to human dermal fibroblasts, with relative viability values close to the control for all samples. Therefore, the incorporation of NPs into the films did not impair cytocompatibility. Chitosan films loaded with gentamicin with or without crosslinking with tannic acid and iron, at the same conditions used in this study, also showed to be nontoxic against human dermal fibroblasts [14].

## 4. Conclusions

In this study, alginate nanoparticles loaded with gentamicin were developed using a simple and reproducible method, based only on the electrostatic interactions between the polymer and the antibiotic. Chitosan-based films loaded with these nanoparticles were produced and compared with previously developed chitosan films, which were loaded with gentamicin. Nanoparticle-loaded films showed a lower swelling degree and a significant decrease of 57% in the gentamicin burst release at pH 7.4. This result could be due to the higher crosslinking degree after nanoparticles incorporation, as noticed by thermal and mechanical analyses. These films also showed pH responsiveness, by releasing 50% more antibiotic at pH 5.5 in comparison to pH 7.4. Moreover, the antibiotic concentration remained above the MIC for an extended period—up to 14 days—indicating that these films can function effectively as wound dressings by sustaining antimicrobial activity until the dressing is changed. The films were also proven to be biocompatible and mechanically robust. The combination of these properties makes these nanoparticle-loaded films potential antibacterial smart dressings, particularly useful for wounds in the initial healing process and low-mobility wound areas.

## Figures and Tables

**Figure 1 polymers-17-02261-f001:**
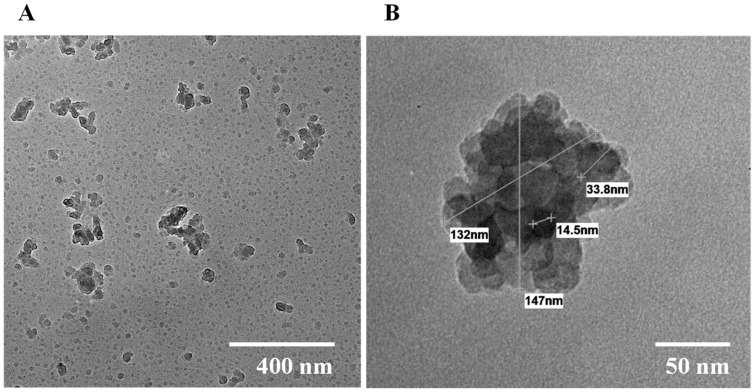
Morphology observed by TEM of alginate–gentamicin nanoparticles prepared with gentamicin at 60% *w*/*w*: (**A**) 8000× magnification; (**B**) 50,000× magnification.

**Figure 2 polymers-17-02261-f002:**
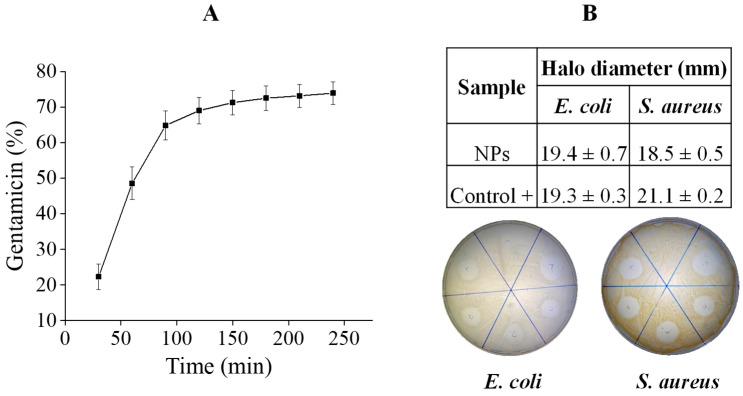
Gentamicin release over time and antimicrobial activity analysis of alginate–gentamicin nanoparticles prepared with gentamicin at 60% *w*/*w*. (**A**) Gentamicin release over time; (**B**) disc diffusion against *E. coli* and *S. aureus*.

**Figure 3 polymers-17-02261-f003:**
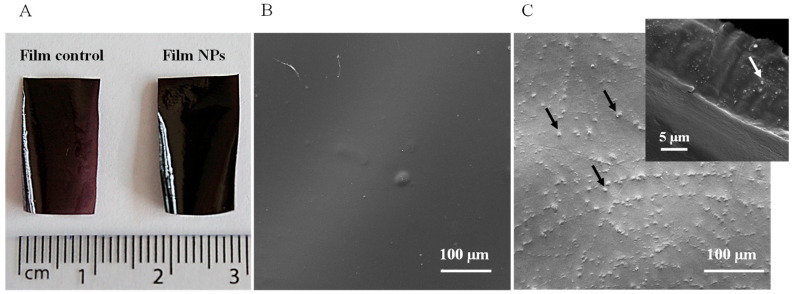
Micrographs of chitosan-based films. (**A**) Film Control and Film NPs. (**B**) Scanning electron microscopy of Film Control. (**C**) Scanning electron microscopy of Film NPs (inset image taken from the transversal section; the arrows indicate the presence of nanoparticles).

**Figure 4 polymers-17-02261-f004:**
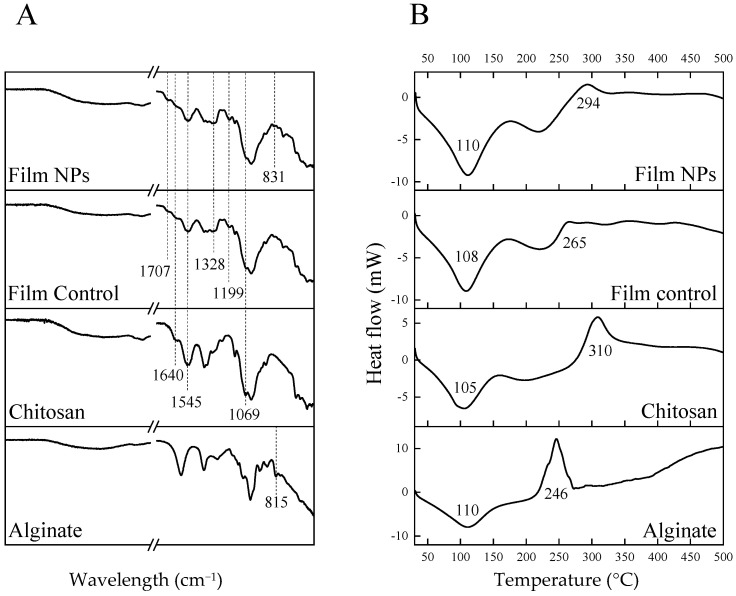
Spectroscopic and thermal analyses of chitosan, alginate and chitosan-based films. (**A**) Fourier transformed infrared spectroscopy-attenuated total reflectance (FTIR-ATR). (**B**) Differential scanning calorimetry (DSC).

**Figure 5 polymers-17-02261-f005:**
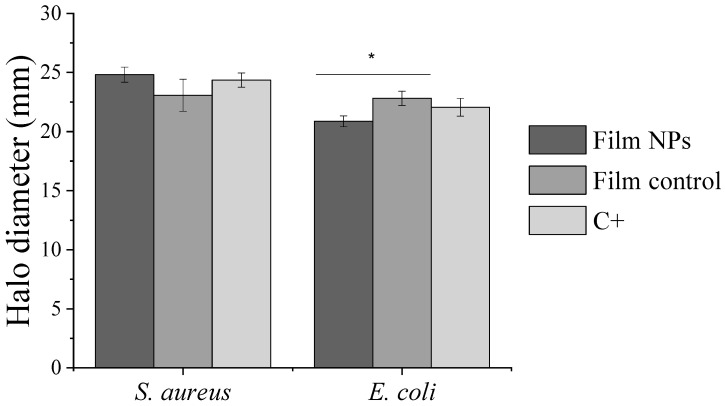
Kirby–Bauer susceptibility test against *S. aureus* and *E. coli* of chitosan-based films loaded with gentamicin and with alginate–gentamicin nanoparticles in comparison to the positive control (C+). * Indicates significant difference between means (*p* < 0.05) according to Tukey’s HSD test.

**Figure 6 polymers-17-02261-f006:**
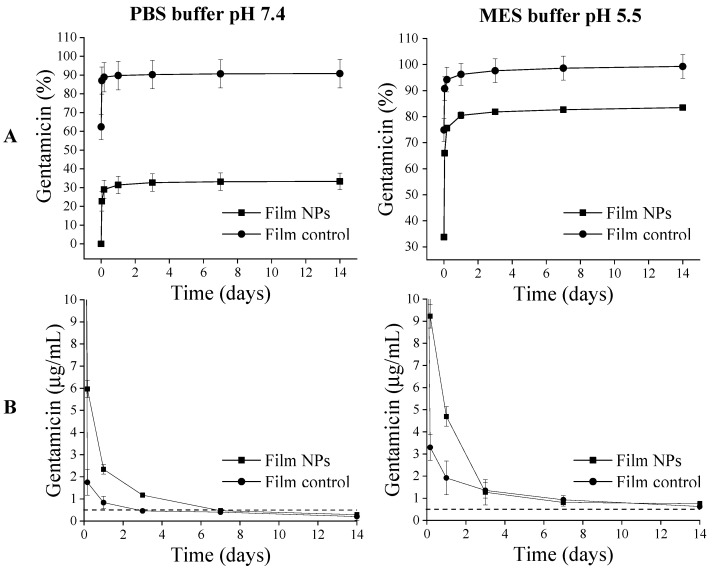
Gentamicin release from chitosan-based films loaded with gentamicin and with alginate–gentamicin nanoparticles at pH 7.4 and pH 5.5. (**A**) Total release of gentamicin over time in %. (**B**) Concentration of gentamicin released per time point in µg/mL. Dotted lines represent the average MIC value of gentamicin against the studied bacteria (*E. coli* and *S. aureus*).

**Figure 7 polymers-17-02261-f007:**
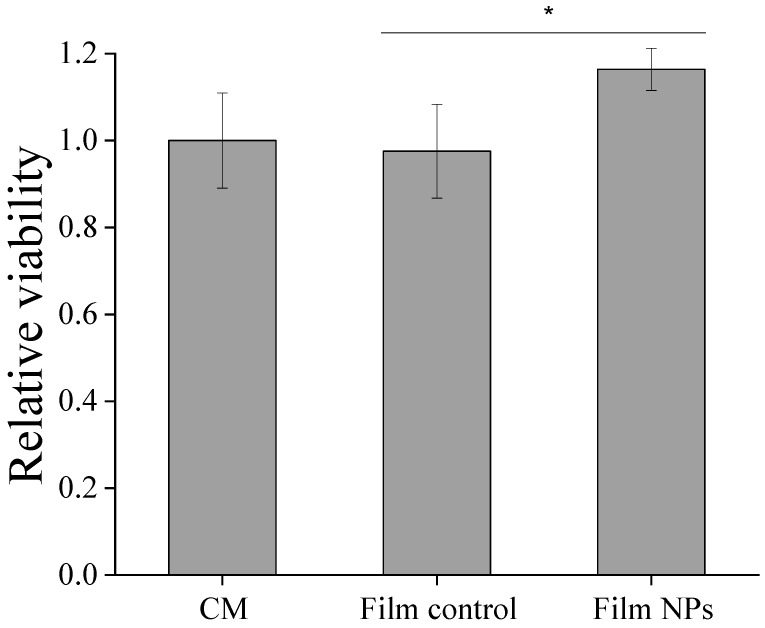
Cytotoxicity against human dermal fibroblasts of chitosan-based films loaded with gentamicin and with alginate–gentamicin nanoparticles. * Indicates significant difference between means (*p* < 0.05) according to Tukey’s HSD test. CM stands for culture media.

**Table 1 polymers-17-02261-t001:** DLS results and gentamicin loading analysis of alginate–gentamicin nanoparticles prepared with different gentamicin percentages in relation to alginate mass.

Gentamicin (% *w*/*w*)	30	40	60
Size (nm)	387.0 ± 138.2 ^a^*	285.2 ± 6.0 ^ab^	86.2 ± 4.4 ^b^
PDI	0.28 ± 0.02 ^a^	0.21 ± 0.02 ^b^	0.15 ± 0.00 ^c^
Zeta (mV)	−44.9 ± 1.2 ^a^	−42.0 ± 0.7 ^b^	−35.0 ± 0.8 ^c^
LC (μg/mg)	296.2 ± 3.2 ^a^	398.6 ± 2.0 ^b^	599.3 ± 1.0 ^c^
LE (%)	98.7 ± 1.1 ^a^	99.6 ± 0.5 ^b^	99.9 ± 0.2 ^b^

Different letters (a, b, c) within a line indicate significant differences between means (*p* < 0.05) according to Tukey’s HSD test. * Bimodal size distribution, in which only the main peak was considered. The original graphs of size distribution and zeta potential are available in Appendix A.

**Table 2 polymers-17-02261-t002:** Thickness, swelling, mass loss and mechanical properties of chitosan-based films loaded with gentamicin and with alginate–gentamicin nanoparticles.

Property	Film Control	Film NPs
Thickness (µm)	14.6 ± 4.2 ^a^	18.4 ± 3.4 ^a^
Swelling (%)	167.3 ± 21.4 ^a^	103.0 ± 11.4 ^b^
Mass loss (%)	25.0 ± 1.6 ^a^	21.9 ± 2.6 ^a^
Tensile strenght (MPa)	32.6 ± 7.9 ^a^	24.6 ± 4.9 ^b^
Elongation at break (%)	1.9 ± 0.9 ^a^	1.5 ± 0.7 ^a^
Young’s modulus (MPa)	38.4 ± 8.4 ^a^	27.6 ± 5.8 ^b^

Different letters (a, b) within a line indicate significant differences between means (*p* < 0.05) according to Tukey’s HSD test.

## Data Availability

Data are available in a publicly accessible repository https://drive.google.com/drive/folders/1tEwmwEw1VWxGpxSL32m5cFFMCJqRHoSG?usp=sharing (accessed on 29 June 2025).

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
