# Peer review of "Chitosan Films Loaded with Alginate Nanoparticles for Gentamicin Release on Demand"

_polymers, 2025, doi:10.3390/polym17162261_

Round 1
Reviewer 1 Report
Comments and Suggestions for Authors
The development of controlled drug release systems is a very important and urgent task of modern medicinal chemistry and polymer science. In this regard, the authors' study on the development of chitosan films loaded with alginate nanoparticles for gentamicin release on-demand is undoubtedly relevant. The novelty of this work lies in the fact that for controlled release, the authors used a new polymer matrix of medium molecular weight chitosan reinforced with alginate nanoparticles. First of all, I would like to note the high scientific level of the study. The text of the article is written competently and logically in good English. In addition, the article is very well illustrated. I will also note that the experimental part is written in sufficient detail and widely uses statistics for processing experimental data. The authors deeply analyze the results and support their conclusions with appropriate references. This article is suitable for the Polymers journal and can be published after revision. The authors should indicate the molecular weight and molecular weight distribution of the polymers used, and the degree of deacetylation of chitosan should also be indicated. In addition, it is necessary to clearly state the hypothesis of this work in the introduction. In addition, the introduction should be rewritten and an emphasis should be placed on the advantages and disadvantages of similar films already developed.
Author Response
REVIEWER 1
Comments and Suggestions for Authors
The development of controlled drug release systems is a very important and urgent task of modern medicinal chemistry and polymer science. In this regard, the authors' study on the development of chitosan films loaded with alginate nanoparticles for gentamicin release on-demand is undoubtedly relevant. The novelty of this work lies in the fact that for controlled release, the authors used a new polymer matrix of medium molecular weight chitosan reinforced with alginate nanoparticles. First of all, I would like to note the high scientific level of the study. The text of the article is written competently and logically in good English. In addition, the article is very well illustrated. I will also note that the experimental part is written in sufficient detail and widely uses statistics for processing experimental data. The authors deeply analyze the results and support their conclusions with appropriate references. This article is suitable for the Polymers journal and can be published after revision. The authors should indicate the molecular weight and molecular weight distribution of the polymers used, and the degree of deacetylation of chitosan should also be indicated. In addition, it is necessary to clearly state the hypothesis of this work in the introduction. In addition, the introduction should be rewritten and an emphasis should be placed on the advantages and disadvantages of similar films already developed.
Answer: Thank you for your comments and suggestions, this helped to improve the quality of the manuscript, we very much appreciated reviewer 1 comments’.
Suitable information available in literature about (i) alginate molecular weight and chitosan’s molecular weight, (ii) molecular weight distribution and (iii) degree of deacetylation were included in the Materials and Methods section, see lines 82-84.
Moreover, the introduction was rewritten as suggested, by stating the hypothesis of the work and including the advantages and disadvantages of similar films in the literature. The modifications can be found in lines 57-78.
Reviewer 2 Report
Comments and Suggestions for Authors
The manuscript presents the development of a wound dressing composed of natural polymers, chitosan and alginate, incorporating gentamicin-loaded nanoparticles, and investigates the pH-responsive antibiotic release properties of this system. The study is comprehensive and well-structured in terms of experimental design, methodology, and results. However, several limitations and suggestions for improvement are outlined below:
- The manuscript references a few prior studies related to pH-responsive systems (e.g., references [24] and [25]). However, I believe a more comprehensive comparative discussion would strengthen the manuscript. Hence, I suggested this in my review.
- The nanoparticle synthesis section reports the use of 30%, 40%, and 60% w/w gentamicin/alginate ratios, but the rationale for selecting this specific range is not explained, nor is there any mention of testing lower or higher values. Therefore, I recommended elaborating on this point.
- The mechanical properties of the films (e.g., flexibility, tensile strength) have not been reported. Given their intended application as wound dressings, mechanical performance is a critical parameter and should be addressed.
- The chitosan was dissolved in 1% acetic acid, which is a commonly used method. However, the potential impact of this solvent on the crystalline structure and hydrophobicity of chitosan was not discussed. I suggested the authors address this point in light of relevant literature that reports structural changes under such conditions.
- The authors claim consistency between TEM and DLS results, but the TEM image shows features that may suggest agglomeration. The authors note that these are likely not artifacts, yet no quantitative analysis is provided. I recommended clarifying this ambiguity with further explanation or quantification.
- Table 1 reports a gentamicin loading efficiency of 99.9%, which is unusually high for a small and hydrophilic antibiotic. Although calibration and validation are briefly mentioned, I believe additional detail is necessary to substantiate this result.
- The release of gentamicin is shown to increase at pH 5.5, and this is attributed to alginate protonation. However, this explanation is somewhat superficial. Given that the pKa values of gentamicin’s amino groups range between 7.15 and 9.2, a more in-depth kinetic discussion of ionization under varying pH conditions would enhance the interpretation.
- The authors should ensure that all abbreviations are introduced only after the full term has been mentioned at least once, typically followed by the acronym in parentheses. This helps avoid confusion and ensures clarity.
Author Response
REVIEWER 2
Comments and Suggestions for Authors
The manuscript presents the development of a wound dressing composed of natural polymers, chitosan and alginate, incorporating gentamicin-loaded nanoparticles, and investigates the pH-responsive antibiotic release properties of this system. The study is comprehensive and well-structured in terms of experimental design, methodology, and results. However, several limitations and suggestions for improvement are outlined below:
1. The manuscript references a few prior studies related to pH-responsive systems (e.g., references [24] and [25]). However, I believe a more comprehensive comparative discussion would strengthen the manuscript. Hence, I suggested this in my review.
Answer (1). Thank you for your appreciation, we valued reviewer 1 comments and suggestions. A comparative discussion was discussed between the authors, and the section was re-written, thus including new references. The main comparison was focused on chitosan-based systems in which the release mechanism is governed by the protonation of amino groups. Also, some examples of different mechanisms were included to enrich the discussion. The changes in the manuscript can be found in lines 492-518.
2. The nanoparticle synthesis section reports the use of 30%, 40%, and 60% w/w gentamicin/alginate ratios, but the rationale for selecting this specific range is not explained, nor is there any mention of testing lower or higher values. Therefore, I recommend elaborating on this point.
Answer (2). Thank you for raising this important point. The rationale for selecting this range was based on previous experiments, in which gentamicin proportions from 10 to 80% were tested. While concentrations below 30% did not enable the formation of nanoparticles, concentrations above 60% formed large precipitates. The rationale was described and justified, see lines 101-102.
3. The mechanical properties of the films (e.g., flexibility, tensile strength) have not been reported. Given their intended application as wound dressings, mechanical performance is a critical parameter and should be addressed.
Question (3). The authors fully agree on the importance of performing mechanical characterization. Therefore, tensile tests were carried out, and the results for tensile strength, elongation at break and Young’s Modulus were included in Table 2. The results were discussed and compared with the literature as can be found in lines 383-394. The authors acknowledge reviewer 2 to underline this point and open the possibility of including additional and complementary mechanical assessment. The manuscript overall quality results are now largely enhanced.
4. The chitosan was dissolved in 1% acetic acid, which is a commonly used method. However, the potential impact of this solvent on the crystalline structure and hydrophobicity of chitosan was not discussed. I suggested the authors address this point in light of relevant literature that reports structural changes under such conditions.
Answer (4). Thank you for the suggestion. When dissolved in acetic acid, chitosan has a semi-crystalline nature, according to the literature (references 24-26). This property was associated with the morphology of the obtained films, as discussed in lines 345-350.
5. The authors claim consistency between TEM and DLS results, but the TEM image shows features that may suggest agglomeration. The authors note that these are likely not artifacts, yet no quantitative analysis is provided. I recommended clarifying this ambiguity with further explanation or quantification.
Answer (5). Thank you for highlighting this point. A deepen analysis was conducted by using the software ImageJ to measure nanoparticles size by TEM. Also, micrography with low magnification was included, to better report the nanoparticles size distribution and correlate with DLS data. The corrections can be found in Figure 1 and lines 316-324.
6. Table 1 reports a gentamicin loading efficiency of 99.9%, which is unusually high for a small and hydrophilic antibiotic. Although calibration and validation are briefly mentioned, I believe additional detail is necessary to substantiate this result.
Answer (6). Thank you for this comment. Indeed, the encapsulation efficiency is unusually high; Nevertheless, all data was validated in triplicates, as mentioned in Materials and Methods. Probably, the electrostatic interactions between alginate and gentamicin are strong enough to lead to high encapsulation values, as well as decrease in size and PDI. This result corroborates the literature, and a new section discussing the point was included in lines 304-307.
7. The release of gentamicin is shown to increase at pH 5.5, and this is attributed to alginate protonation. However, this explanation is somewhat superficial. Given that the pKa values of gentamicin’s amino groups range between 7.15 and 9.2, a more in-depth kinetic discussion of ionization under varying pH conditions would enhance the interpretation.
Answer (7). Thank you for the suggestion. Gentamicin release is dependent on both chitosan, alginate and gentamicin protonation states in different pHs. An improved discussion about gentamicin release behavior and a correlation with pKa values was included, as added in lines 468-477.
8. The authors should ensure that all abbreviations are introduced only after the full term has been mentioned at least once, typically followed by the acronym in parentheses. This helps avoid confusion and ensures clarity.
Answer (8). Thank you for the comment. All the abbreviations were double checked and corrected.
Reviewer 3 Report
Comments and Suggestions for Authors
In this manuscript, the author evaluated gentamicin release from chitosan films loaded with alginate nanoparticles, high loading efficiency and pH-responsive delivery system was developed for the treatment of skin wounds. In addition, the whole manuscript was in a relative good organizing and writing, the following issues should be addressed before acceptance:
- There are many reported natural polymers drug-loaded delivery system, what is the novelty of this work? Which should emphasized in the section of Introduction.
- The picture of the obtained composites should provided.
- For the Size and Zeta potential results, representative distribution curve should provided.
- Line 250-252, to demonstrate the “strong electrostatic interaction”, FTIR spectra should provided.
- The caption of Figure 4 should revised, the dotted curves and square-dot curves are not indicated to which drug system they belong respectively.
- For the gentamicin total release, why does the dotted curves present similar release behavior at different pH values? Why does the square-dot curves present different release behavior? What is the reason?
- How is the thermal and mechanical properties of the obtained composite films?

Author Response
REVIEWER 3
Comments and Suggestions for Authors
In this manuscript, the author evaluated gentamicin release from chitosan films loaded with alginate nanoparticles, high loading efficiency and pH-responsive delivery system was developed for the treatment of skin wounds. In addition, the whole manuscript was in a relative good organizing and writing, the following issues should be addressed before acceptance:
1. There are many reported natural polymers drug-loaded delivery system, what is the novelty of this work? Which should emphasized in the section of Introduction.
Answer (1). Thank you for this comment. The novelty of the work was emphasized in the Introduction section - as suggested. This work stressed the emerging approach of combining drug-loaded nanoparticles and polymeric matrices to control drug release and obtain a pH-responsive system. To the best of our knowledge, this is one of the first studies to develop and characterize pH-sensitive composite system using alginate-chitosan combinations for the localized delivery of gentamicin. A new section describing the originality, the pertinence and the impact of the approach was added – see lines 57-78.
2. The picture of the obtained composites should provided.
Answer (2). Thank you for this suggestion. The macrography of the films was provided in Figure 3, and the discussion on morphology and aspect was improved as can be seen in lines 345-358.
3. For the Size and Zeta potential results, representative distribution curve should provided.
Answer (3). The original graphs for size distribution by number and Zeta potential values obtained by DLS were provided in the Supplementary Materials, following your valuable suggestion.
4. Line 250-252, to demonstrate the “strong electrostatic interaction”, FTIR spectra should provided.
Answer (4). To deeply understand the interactions among the film’s components, FTIR-ATR and DSC analysis were carried out. FTIR-ATR analysis resulted in subtle changes in the spectra for Film NPs compared to the Film control and it was not possible to draw firm conclusions. On the other hand, DSC analysis was more conclusive, allowing inferences regarding the interaction between chitosan and alginate. The results and discussion for FTIR-ATR and DSC can now be visualized in lines 395-428.
5. The caption of Figure 4 should revised, the dotted curves and square-dot curves are not indicated to which drug system they belong respectively.
Answer (5). The authors appreciated this observation. The legend in Figure 4 has been improved for better readability and clarity.
6. For the gentamicin total release, why does the dotted curves present similar release behavior at different pH values? Why does the square-dot curves present different release behavior? What is the reason?
Answer (6). Thank you for the comment. The discussion on the different behaviors of the samples in different pHs was rewritten and improved for better understanding. The revised text was added and can be seen in lines 448-477.
7. How is the thermal and mechanical properties of the obtained composite films?
Answer (7). The samples were further characterized for mechanical properties by tensile tests, and thermal properties by DSC analysis. These analyses enabled a significant improvement in the manuscript, and new conclusions could be drawn. The revised text is presented in lines 383-428.
Round 2
Reviewer 2 Report
Comments and Suggestions for Authors
The authors’ responses are generally adequate and satisfactory. They have provided detailed, literature-supported answers to each critique and have made the necessary revisions and additions, thereby improving the quality of the manuscript.
Reviewer 3 Report
Comments and Suggestions for Authors
Corrections were performed, the manuscript has been improved and now it is suitable for publishing